# Deep Residual Learning in Spiking Neural Networks

Wei Fang[1,2], Zhaofei Yu[1,2*], Yanqi Chen[1,2],

Tiejun Huang[1,2], Timothée Masquelier[3], Yonghong Tian[1,2*]

[1]Department of Computer Science and Technology, Peking University
[2]Peng Cheng Laboratory, Shenzhen 518055, China
[3]Centre de Recherche Cerveau et Cognition, UMR5549 CNRS - Univ. Toulouse 3 , Toulouse, France

## Abstract

Deep Spiking Neural Networks (SNNs) present optimization difficulties for gradient-based approaches due to discrete binary activation and complex spatial-temporal dynamics. Considering the huge success of ResNet in deep learning, it would be natural to train deep SNNs with residual learning. Previous Spiking ResNet mimics the standard residual block in ANNs and simply replaces ReLU activation layers with spiking neurons, which suffers the degradation problem and can hardly implement residual learning. In this paper, we propose the spike-element-wise (SEW) ResNet to realize residual learning in deep SNNs. We prove that the SEW ResNet can easily implement identity mapping and overcome the vanishing/exploding gradient problems of Spiking ResNet. We evaluate our SEW ResNet on ImageNet, DVS Gesture, and CIFAR10-DVS datasets, and show that SEW ResNet outperforms the state-of-the-art directly trained SNNs in both accuracy and time-steps. Moreover, SEW ResNet can achieve higher performance by simply adding more layers, providing a simple method to train deep SNNs. To our best knowledge, this is the first time that directly training deep SNNs with more than 100 layers becomes possible. Our codes are available at `https://github.com/fangwei123456/Spike-Element-Wise-ResNet`.

## 1 Introduction

Artificial Neural Networks (ANNs) have achieved great success in many tasks, including image classification [28, 52, 55], object detection [9, 34, 44], machine translation [2], and gaming [37, 51]. One of the critical factors for ANNs' success is deep learning [29], which uses multi-layers to learn representations of data with multiple levels of abstraction. It has been proved that deeper networks have advantages over shallower networks in computation cost and generalization ability [3]. The function represented by a deep network can require an exponential number of hidden units by a shallow network with one hidden layer [38]. In addition, the depth of the network is closely related to the network's performance in practical tasks [52, 55, 27, 52]. Nevertheless, recent evidence [13, 53, 14] reveals that with the network depth increasing, the accuracy gets saturated and then degrades rapidly. To solve this degradation problem, residual learning is proposed [14, 15] and the residual structure is widely exploited in "very deep" networks that achieve the leading performance [22, 59, 18, 57].

Spiking Neural Networks (SNNs) are regarded as a potential competitor of ANNs for their high biological plausibility, event-driven property, and low power consumption [45]. Recently, deep learning methods are introduced into SNNs, and deep SNNs have achieved close performance as ANNs in some simple classification datasets [56], but still worse than ANNs in complex tasks, e.g., classifying the ImageNet dataset [47]. To obtain higher performance SNNs, it would be natural to

---

[*]Corresponding author

35th Conference on Neural Information Processing Systems (NeurIPS 2021).

explore deeper network structures like ResNet. Spiking ResNet [25, 60, 21, 17, 49, 12, 30, 64, 48, 42, 43], as the spiking version of ResNet, is proposed by mimicking the residual block in ANNs and replacing ReLU activation layers with spiking neurons. Spiking ResNet converted from ANN achieves state-of-the-art accuracy on nearly all datasets, while the directly trained Spiking ResNet has not been validated to solve the degradation problem.

In this paper, we show that Spiking ResNet is inapplicable to all neuron models to achieve identity mapping. Even if the identity mapping condition is met, Spiking ResNet suffers from the problems of vanishing/exploding gradient. Thus, we propose the Spike-Element-Wise (SEW) ResNet to realize residual learning in SNNs. We prove that the SEW ResNet can easily implement identity mapping and overcome the vanishing/exploding gradient problems at the same time. We evaluate Spiking ResNet and SEW ResNet on both the static ImageNet dataset and the neuromorphic DVS Gesture dataset [1], CIFAR10-DVS dataset [32]. The experiment results are consistent with our analysis, indicating that the deeper Spiking ResNet suffers from the degradation problem — the deeper network has higher training loss than the shallower network, while SEW ResNet can achieve higher performance by simply increasing the network's depth. Moreover, we show that SEW ResNet outperforms the state-of-the-art directly trained SNNs in both accuracy and time-steps. To the best of our knowledge, this is the first time to explore the directly-trained deep SNNs with more than 100 layers.

## 2 Related Work

### 2.1 Learning Methods of Spiking Neural Networks

ANN to SNN conversion (ANN2SNN) [20, 4, 46, 49, 12, 11, 6, 54, 33] and backpropagation with surrogate gradient [40] are the two main methods to get deep SNNs. The ANN2SNN method firstly trains an ANN with ReLU activation, then converts the ANN to an SNN by replacing ReLU with spiking neurons and adding scaling operations like weight normalization and threshold balancing. Some recent conversion methods have achieved near loss-less accuracy with VGG-16 and ResNet [12, 11, 6, 33]. However, the converted SNN needs a longer time to rival the original ANN in precision as the conversion is based on rate-coding [46], which increases the SNN's latency and restricts the practical application. The backpropagation methods can be classified into two categories [26]. The method in the first category computes the gradient by unfolding the network over the simulation time-steps [31, 19, 58, 50, 30, 40], which is similar to the idea of backpropagation through time (BPTT). As the gradient with respect to the threshold-triggered firing is non-differentiable, the surrogate gradient is often used. The SNN trained by the surrogate method is not limited to rate-coding, and can also be applied on temporal tasks, e.g., classifying neuromorphic datasets [58, 8, 16]. The second method computes the gradients of the timings of existing spikes with respect to the membrane potential at the spike timing [5, 39, 24, 65, 63].

### 2.2 Spiking Residual Structure

Previous ANN2SNN methods noticed the distinction between plain feedforward ANNs and residual ANNs, and made specific normalization for conversion. Hu et al. [17] were the first to apply the residual structure in ANN2SNN with scaled shortcuts in SNN to match the activations of the original ANN. Sengupta et al. [49] proposed Spike-Norm to balance SNN's threshold and verified their method by converting VGG and ResNet to SNNs. Existing backpropagation-based methods use nearly the same structure from ResNet. Lee et al. [30] evaluated their custom surrogate methods on shallow ResNets whose depths are no more than ResNet-11. Zheng et al. [64] proposed the threshold-dependent batch normalization (td-BN) to replace naive batch normalization (BN) [23] and successfully trained Spiking ResNet-34 and Spiking ResNet-50 directly with surrogate gradient by adding td-BN in shortcuts.

## 3 Methods

### 3.1 Spiking Neuron Model

The spiking neuron is the fundamental computing unit of SNNs. Similar to Fang et al. [8], we use a unified model to describe the dynamics of all kinds of spiking neurons, which includes the following

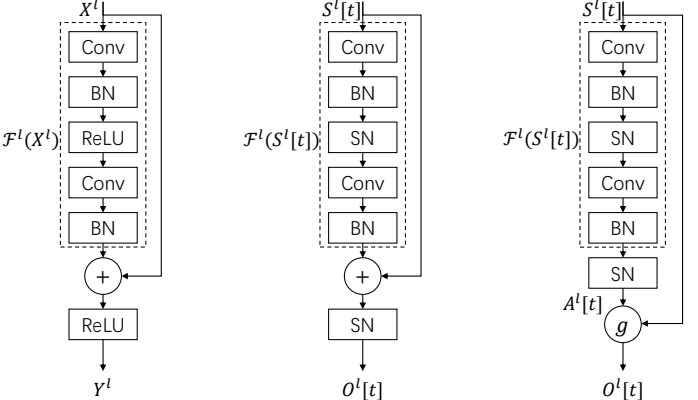

(a) Basic block in ResNet    (b) Basic block in Spiking ResNet  (c) Spike-Element-Wise block

Figure 1: Residual blocks in ResNet, Spiking ResNet and SEW ResNet.

discrete-time equations:

$$H[t] = f(V[t-1], X[t]), \tag{1}$$

$$S[t] = \Theta(H[t] - V_{th}), \tag{2}$$

$$V[t] = H[t] \ (1 - S[t]) + V_{reset} \ S[t], \tag{3}$$

where $X[t]$ is the input current at time-step $t$, $H[t]$ and $V[t]$ denote the membrane potential after neuronal dynamics and after the trigger of a spike at time-step $t$, respectively. $V_{th}$ is the firing threshold, $\Theta(x)$ is the Heaviside step function and is defined by $\Theta(x) = 1$ for $x \geq 0$ and $\Theta(x) = 0$ for $x < 0$. $S[t]$ is the output spike at time-step $t$, which equals 1 if there is a spike and 0 otherwise. $V_{reset}$ denotes the reset potential. The function $f(\cdot)$ in Eq. (1) describes the neuronal dynamics and takes different forms for different spiking neuron models. For example, the function $f(\cdot)$ for the Integrate-and-Fire (IF) model and Leaky Integrate-and-Fire (LIF) model can be described by Eq. (4) and Eq. (5), respectively.

$$H[t] = V[t-1] + X[t], \tag{4}$$

$$H[t] = V[t-1] + \frac{1}{\tau}(X[t] - (V[t-1] - V_{reset})), \tag{5}$$

where $\tau$ represents the membrane time constant. Eq. (2) and Eq. (3) describe the spike generation and resetting processes, which are the same for all kinds of spiking neuron models. In this paper, the surrogate gradient method is used to define $\Theta'(x) \triangleq \sigma'(x)$ during error back-propagation, with $\sigma(x)$ denoting the surrogate function.

## 3.2 Drawbacks of Spiking ResNet

The residual block is the key component of ResNet. Fig. 1(a) shows the basic block in ResNet [14], where $X^l, Y^l$ are the input and output of the $l$-th block in ResNet, Conv is the convolutional layer, BN denotes batch normalization, and ReLU denotes the rectified linear unit activation layer. The basic block of Spiking ResNet used in [64, 17, 30] simply mimics the block in ANNs by replacing ReLU activation layers with spiking neurons (SN), which is illustrated in Fig. 1(b). Here $S^l[t], O^l[t]$ are the input and output of the $l$-th block in Spiking ResNet at time-step $t$. Based on the above definition, we will analyze the drawbacks of Spiking ResNet below.

**Spiking ResNet is inapplicable to all neuron models to achieve identity mapping.** One of the critical concepts in ResNet is identity mapping. He et al. [14] noted that if the added layers implement the identity mapping, a deeper model should have training error no greater than its shallower counterpart. However, it is unable to train the added layers to implement identity mapping in a feasible time, resulting in deeper models performing worse than shallower models (the degradation problem). To solve this problem, the residual learning is proposed by adding a shortcut connection (shown in Fig. 1(a)). If we use $\mathcal{F}^l$ to denote the residual mapping, e.g., a stack of two convolutional layers, of the $l$-th residual block in ResNet and Spiking ResNet, then the residual block in Fig.1(a)

and Fig.1(b) can be formulated as

$$Y^l = \text{ReLU}(\mathcal{F}^l(X^l) + X^l), \tag{6}$$

$$O^l[t] = \text{SN}(\mathcal{F}^l(S^l[t]) + S^l[t]). \tag{7}$$

The residual block of Eq. (6) make it easy to implement identity mapping in ANNs. To see this, when $\mathcal{F}^l(X^l) \equiv 0$, $Y^l = \text{ReLU}(X^1)$. In most cases, $X^l$ is the activation of the previous ReLU layer and $X^l \geq 0$. Thus, $Y^l = \text{ReLU}(X^l) = X^l$, which is identity mapping.

Different from ResNet, the residual block in Spiking ResNet (Eq. (7)) restricts the models of spiking neuron to implement identity mapping. When $\mathcal{F}^l(S^l[t]) \equiv 0$, $O^l[t] = \text{SN}(S^l[t]) \neq S^l[t]$. To transmit $S^l[t]$ and make $\text{SN}(S^l[t]) = S^l[t]$, the last spiking neuron (SN) in the $l$-th residual block needs to fire a spike after receiving a spike, and keep silent after receiving no spike at time-step $t$. It works for IF neuron described by Eq. (4). Specifically, we can set $0 < V_{th} \leq 1$ and $V[t-1] = 0$ to ensure that $X[t] = 1$ leads to $H[t] \geq V_{th}$, and $X[t] = 0$ leads to $H[t] < V_{th}$. However, when considering some spiking neuron models with complex neuronal dynamics, it is hard to achieve $\text{SN}(S^l[t]) = S^l[t]$. For example, the LIF neuron used in [66, 8, 61] considers a learnable membrane time constant $\tau$, the neuronal dynamics of which can be described with Eq. (5). When $X[t] = 1$ and $V[t-1] = 0$, $H[t] = \frac{1}{\tau}$. It is difficult to find a firing threshold that ensures $H[t] > V_{th}$ as $\tau$ is being changed in training by the optimizer.

**Spiking ResNet suffers from the problems of vanishing/exploding gradient.** Consider a spiking ResNet with $k$ sequential blocks to transmit $S^l[t]$, and the identity mapping condition is met, e.g., the spiking neurons are the IF neurons with $0 < V_{th} \leq 1$, then we have $S^l[t] = S^{l+1}[t] = ... = S^{l+k-1}[t] = O^{l+k-1}[t]$. Denote the $j$-th element in $S^l[t]$ and $O^l[t]$ as $S^l_j[t]$ and $O^l_j[t]$ respectively, the gradient of the output of the $(l + k - 1)$-th residual block with respect to the input of the $l$-th residual block can be calculated layer by layer:

$$\frac{\partial O^{l+k-1}_j[t]}{\partial S^l_j[t]} = \prod_{i=0}^{k-1} \frac{\partial O^{l+i}_j[t]}{\partial S^{l+i}_j[t]} = \prod_{i=0}^{k-1} \Theta'(S^{l+i}_j[t] - V_{th}) \to \begin{cases} 0, \text{if } 0 < \Theta'(S^l_j[t] - V_{th}) < 1 \\ 1, \text{if } \Theta'(S^l_j[t] - V_{th}) = 1 \\ +\infty, \text{if } \Theta'(S^l_j[t] - V_{th}) > 1 \end{cases}, \tag{8}$$

where $\Theta(x)$ is the Heaviside step function and $\Theta'(x)$ is defined by the surrogate gradient. The second equality hold as $O^{l+i}_j[t] = \text{SN}(S^{l+i}_j[t])$. In view of the fact that $S^l_j[t]$ can only take 0 or 1, $\Theta'(S^l_j[t] - V_{th}) = 1$ is not satisfied for commonly used surrogate functions mentioned in [40]. Thus, the vanishing/exploding gradient problems are prone to happen in deeper Spiking ResNet.

Based on the above analysis, we believe that the previous Spiking ResNet ignores the highly nonlinear caused by spiking neurons, and can hardly implement residual learning. Nonetheless, the basic block in Fig. 1(b) is still decent for ANN2SNN with extra normalization [17, 49], as the SNN converted from ANN aims to use firing rates to match the origin ANN's activations.

### 3.3 Spike-Element-Wise ResNet

Here we propose the Spike-Element-Wise (SEW) residual block to realize the residual learning in SNNs, which can easily implement identity mapping and overcome the vanishing/exploding gradient problems at the same time. As illustrated in Fig. 1(c), the SEW residual block can be formulated as:

$$O^l[t] = g(\text{SN}(\mathcal{F}^l(S^l[t])), S^l[t]) = g(A^l[t], S^l[t]), \tag{9}$$

where $g$ represents an element-wise function with two spikes tensor as inputs. Here we use $A^l[t]$ to denote the residual mapping to be learned as $A^l[t] = \text{SN}(\mathcal{F}^l(S^l[t]))$.

**SEW ResNet can easily implement identity mapping.** By utilizing the binary property of spikes, we can find different element-wise functions $g$ that satisfy identity mapping (shown in Tab. 1). To be specific, when choosing *ADD* and *IAND* as element-wise functions $g$, identity mapping is achieved by setting $A^l[t] \equiv 0$, which can be implemented simply by setting

| Name | Expression of $g(A^l[t], S^l[t])$ |
|------|-----------------------------------|
| ADD | $A^l[t] + S^l[t]$ |
| AND | $A^l[t] \wedge S^l[t] = A^l[t] \cdot S^l[t]$ |
| IAND | $(\neg A^l[t]) \wedge S^l[t] = (1 - A^l[t]) \cdot S^l[t]$ |

Table 1: List of element-wise functions $g$.

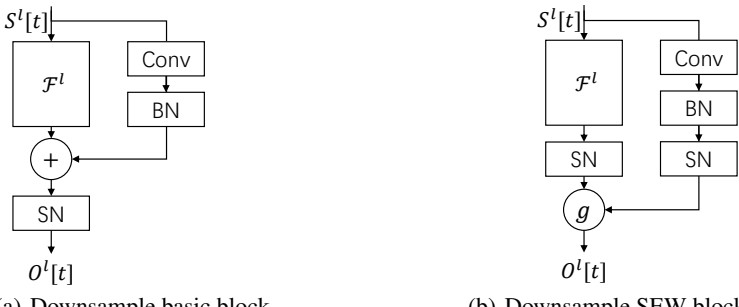

(a) Downsample basic block          (b) Downsample SEW block

Figure 2: Downsample blocks in Spiking ResNet and SEW ResNet.

the weights and the bias of the last batch normalization layer (BN) in $\mathcal{F}^l$ to zero. Then we can get $O^l[t] = g(A^l[t], S^l[t]) = g(\mathrm{SN}(0), S^l[t]) = g(0, S^l[t]) = S^l[t]$. This is applicable to all neuron models. When using *AND* as the element-wise function $g$, we set $A^l[t] \equiv 1$ to get identity mapping. It can be implemented by setting the last BN's weights to zero and the bias to a large enough constant to cause spikes, e.g., setting the bias as $V_{th}$ when the last SN is IF neurons. Then we have $O^l[t] = 1 \wedge S^l[t] = S^l[t]$. Note that using *AND* may suffer from the same problem as Spiking ResNet. It is hard to control some spiking neuron models with complex neuronal dynamics to generate spikes at a specified time-step.

**Formulation of downsample block.** Remarkably, when the input and output of one block have different dimensions, the shortcut is set as convolutional layers with stride $> 1$, rather than the identity connection, to perform downsampling. The ResNet and the Spiking ResNet utilize {Conv-BN} without ReLU in shortcut (Fig. 2(a)). In contrast, we add a SN in shortcut (Fig. 2(b)).

**SEW ResNet can overcome vanishing/exploding gradient.** The SEW block is similar to *ReLU before addition* (RBA) block [15] in ANNs, which can be formulated as

$$Y^l = \mathrm{ReLU}(\mathcal{F}^l(X^l)) + X^l. \tag{10}$$

The RBA block is criticized by He et al. [15] for $X^{l+1} = Y^l \geq X^l$, which will cause infinite outputs in deep layers. The experiment results in [15] also showed that the performance of the RBA block is worse than the basic block (Fig.1(a)). To some extent, the SEW block is an extension of the RBA block. Note that using *AND* and *IAND* as $g$ will output spikes (i.e. binary tensors), which means that the infinite outputs problem in ANNs will never occur in SNNs with SEW blocks, since all spikes are less or equal than 1. When choosing *ADD* as $g$, the infinite outputs problem can be relieved as the output of $k$ sequential SEW blocks will be no larger than $k + 1$. In addition, a downsample SEW block will regulate the output to be no larger than 2 when $g$ is *ADD*.

When the identity mapping is implemented, the gradient of the output of the $(l + k - 1)$-th SEW block with respect to the input of the $l$-th SEW block can be calculated layer by layer:

$$\frac{\partial O_j^{l+k-1}[t]}{\partial S_j^l[t]} = \prod_{i=0}^{k-1} \frac{\partial g(A_j^{l+i}[t], S_j^{l+i}[t])}{\partial S_j^{l+i}[t]} = \begin{cases} \prod_{i=0}^{k-1} \frac{\partial (0 + S_j^{l+i}[t])}{\partial S_j^{l+i}[t]}, \text{if } g = ADD \\ \prod_{i=0}^{k-1} \frac{\partial (1 \cdot S_j^{l+i}[t])}{\partial S_j^{l+i}[t]}, \text{if } g = AND \\ \prod_{i=0}^{k-1} \frac{\partial ((1-0) \cdot S_j^{l+i}[t])}{\partial S_j^{l+i}[t]}, \text{if } g = IAND \end{cases} = 1. \tag{11}$$

The second equality holds as identity mapping is achieved by setting $A^{l+i}[t] \equiv 1$ for $g = AND$, and $A^{l+i}[t] \equiv 0$ for $g = ADD/IAND$. Since the gradient in Eq. (11) is a constant, the SEW ResNet can overcome the vanishing/exploding gradient problems.

## 4 Experiments

### 4.1 ImageNet Classification

As the test server of ImageNet 2012 is no longer available, we can not report the actual test accuracy. Instead, we use the accuracy on the *validation* set as the test accuracy, which is the same as [17, 64].

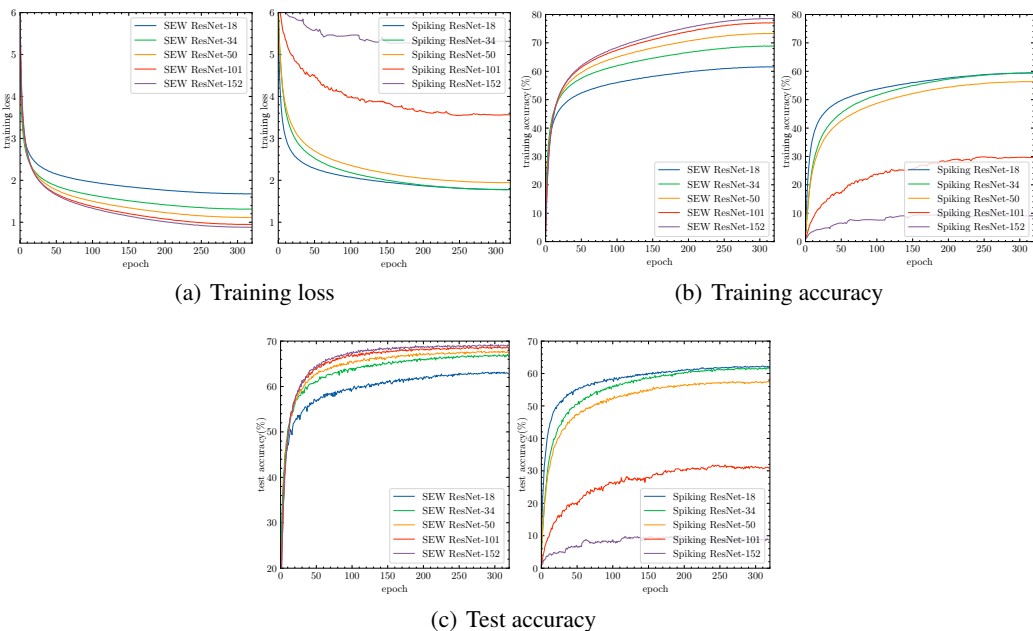

(a) Training loss

(b) Training accuracy

(c) Test accuracy

Figure 3: Comparison of the training loss, training accuracy and test accuracy on ImageNet.

| Network | SEW ResNet (ADD) | | Spiking ResNet | |
|---|---|---|---|---|
| | Acc@1(%) | Acc@5(%) | Acc@1(%) | Acc@5(%) |
| ResNet-18 | 63.18 | 84.53 | 62.32 | 84.05 |
| ResNet-34 | 67.04 | 87.25 | 61.86 | 83.69 |
| ResNet-50 | 67.78 | 87.52 | 57.66 | 80.43 |
| ResNet-101 | 68.76 | 88.25 | 31.79 | 54.91 |
| ResNet-152 | 69.26 | 88.57 | 10.03 | 23.57 |

Table 2: Test accuracy on ImageNet.

He et al. [14] evaluated the 18/34/50/101/152-layer ResNets on the ImageNet dataset. For comparison, we consider the SNNs with the same network architectures, except that the basic residual block (Fig.1(a)) is replaced by the spiking basic block (Fig.1(b)) and SEW block (Fig.1(c)) with $g$ as *ADD*, respectively. We denote the SNN with the basic block as *Spiking ResNet* and the SNN with the SEW block as *SEW ResNet*. The IF neuron model is adopted for the static ImageNet dataset. During training on ImageNet, we find that the Spiking ResNet-50/101/152 can not converge unless we use the zero initialization [10], which sets all blocks to be an identity mapping at the start of training. Thus, the results of Spiking ResNet-18/34/50/101/152 reported in this paper are with zero initialization.

**Spiking ResNet *vs.* SEW ResNet.** We first evaluate the performance of Spiking ResNet and SEW ResNet. Tab. 2 reports the test accuracy on ImageNet validation. The results show that the deeper 34-layer Spiking ResNet has lower test accuracy than the shallower 18-layer Spiking ResNet. As the layer increases, the test accuracy of Spiking ResNet decreases. To reveal the reason, we compare the training loss, training accuracy, and test accuracy of Spiking ResNet during the training procedure, which is shown in Fig. 3. We can find the degradation problem of the Spiking ResNet — the deeper network has higher *training loss* than the shallower network. In contrast, the deeper 34-layer SEW ResNet has higher test accuracy than the shallower 18-layer SEW ResNet (shown in Tab. 2). More importantly, it can be found from Fig. 3 that the training loss of our SEW ResNet decreases and the training/test accuracy increases with the increase of depth, which indicates that we can obtain higher performance by simply increasing the network's depth. All these results imply that the degradation problem is well addressed by SEW ResNet.

**Comparisons with State-of-the-art Methods.** In Tab. 3, we compare SEW ResNet with previous Spiking ResNets that achieve the best results on ImageNet. To our best knowledge, the SEW ResNet-101 and the SEW ResNet-152 are the only SNNs with more than 100 layers to date, and there are no other networks with the same structure to compare. When the network structure is the same, our SEW ResNet outperforms the state-of-the-art accuracy of directly trained Spiking ResNet, even with fewer

| Network | Methods | Accuracy(%) | T |
|---|---|---|---|
| **SEW ResNet-34** | **Spike-based BP** | **67.04** | **4** |
| Spiking ResNet-34(large)† with td-BN [64] | Spike-based BP | 67.05 | 6 |
| Spiking ResNet-34 with td-BN [64] | Spike-based BP | 63.72 | 6 |
| Spiking ResNet-34 [12] | ANN2SNN | 69.89 | 4096 |
| Spiking ResNet-34 [49] | ANN2SNN | 65.47 | 2000 |
| Spiking ResNet-34 [33] | ANN2SNN | 74.61 | 256 |
| Spiking ResNet-34 [43] | ANN2SNN and Spike-based BP | 61.48 | 250 |
| **SEW ResNet-50** | **Spike-based BP** | **67.78** | **4** |
| Spiking ResNet-50 with td-BN [64] | Spike-based BP | 64.88 | 6 |
| Spiking ResNet-50 [17] | ANN2SNN | 72.75 | 350 |
| **SEW ResNet-101** | **Spike-based BP** | **68.76** | **4** |
| **SEW ResNet-152** | **Spike-based BP** | **69.26** | **4** |

Table 3: Comparison with previous Spiking ResNet on ImageNet. † has the same network structure as the standard Spiking ResNet-34, but uses four times as many the number of convolution kernels.

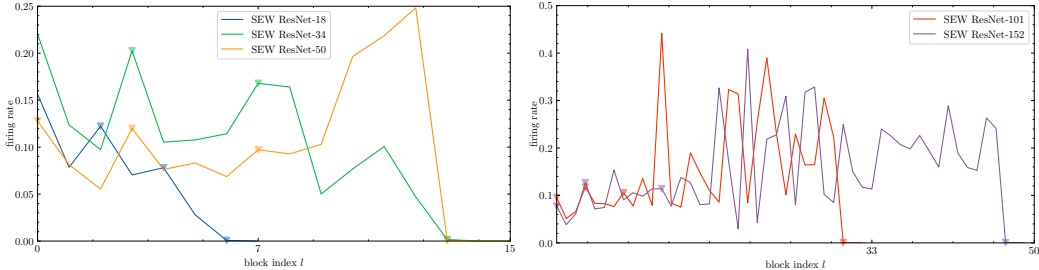

Figure 4: Firing rates of $A^l$ in SEW blocks on ImageNet.

time-steps $T$. The accuracy of SEW ResNet-34 is slightly lower than Spiking ResNet-34 (large) with td-BN (67.04% v.s. 67.05%), which uses 1.5 times as many simulating time-steps $T$ (6 v.s. 4) and 4 times as many the number of parameters (85.5M v.s. 21.8M), compared with our SEW ResNet. The state-of-the-art ANN2SNN methods [33, 17] have better accuracy than our SEW ResNet, but they respectively use 64 and 87.5 times as many time-steps as ours.

**Analysis of spiking response of SEW blocks.** Fig. 4 shows the firing rates of $A^l$ in SEW ResNet-18/34/50/101/152 on ImageNet. There are 7 blocks in SEW ResNet-18, 15 blocks in SEW ResNet-34 and SEW ResNet-50, 33 blocks in SEW ResNet-101, and 50 blocks in SEW ResNet-152. The downsample SEW blocks are marked by the triangle down symbol ▽. As we choose *ADD* as element-wise functions $g$, a lower firing rate means that the SEW block gets closer to implementing identity mapping, except for downsample blocks. Note that the shortcuts of downsample blocks are not identity mapping, which is illustrated in Fig. 2(b). Fig. 4 shows that all spiking neurons in SEW blocks have low firing rates, and the spiking neurons in the last two blocks even have firing rates of almost zero. As the time-steps $T$ is 4 and firing rates are no larger than 0.25, all neurons in SEW ResNet-18/34/50 fire on average no more than one spike during the whole simulation. Besides, all firing rates in SEW ResNet-101/152 are not larger than 0.5, indicating that all neurons fire on average not more than two spikes. In general, the firing rates of $A^l$ in SEW blocks are at a low level, verifying that most SEW blocks act as identity mapping.

**Gradients Check on ResNet-152 Structure.** Eq. (8) and Eq. (11) analyze the gradients of multiple blocks with identity mapping. To verify that SEW ResNet can overcome vanishing/exploding gradient, we check the gradients of Spiking ResNet-152 and SEW ResNet-152, which are the deepest standard ResNet structure. We consider the same initialization parameters and with/without zero initialization.

As the gradients of SNNs are significantly influenced by firing rates (see Sec.A.4), we analyze the firing rate firstly. Fig. 5(a) shows the initial firing rate of $l$-th block's output $O^l$. The indexes of downsample blocks are marked by vertical dotted lines. The blocks between two adjacent dotted lines represent the identity mapping areas, and have inputs and outputs with the same shape. When using zero initialization, Spiking ResNet, SEW AND ResNet, SEW IAND ResNet, and SEW ADD ResNet have the same firing rates (green curve), which is the *zero init* curve. Without zero initialization, the silence problem happens in the SEW AND network (red curve), and is relieved by the SEW IAND network (purple curve). Fig. 5(b) shows the firing rate of $A^l$, which represents the output of last SN in $l$-th block. It can be found that although the firing rate of $O^l$ in SEW ADD ResNet increases linearly in the identity mapping areas, the last SN in each block still maintains a stable firing rate. Note that when $g$ is *ADD*, the output of the SEW block is not binary, and the firing

rate is actually the mean value. The SNs of SEW IAND ResNet maintain an adequate firing rate and decay slightly with depth (purple curve), while SNs in deep layers of SEW AND ResNet keep silent (orange curve). The silence problem can be explained as follows. When using *AND*, $O^l[t] = \text{SN}(\mathcal{F}^l(O^{l-1}[t])) \wedge O^{l-1}[t] \le O^{l-1}[t]$. Since it is hard to keep $\text{SN}(\mathcal{F}^l(O^{l-1}[t])) \equiv 1$ at each time-step $t$, the silence problem may frequently happen in SEW ResNet with *AND* as $g$. Using *IAND* as a substitute of *AND* can relieve this problem because it is easy to keep $\text{SN}(\mathcal{F}^l(O^{l-1}[t])) \equiv 0$ at each time-step $t$.

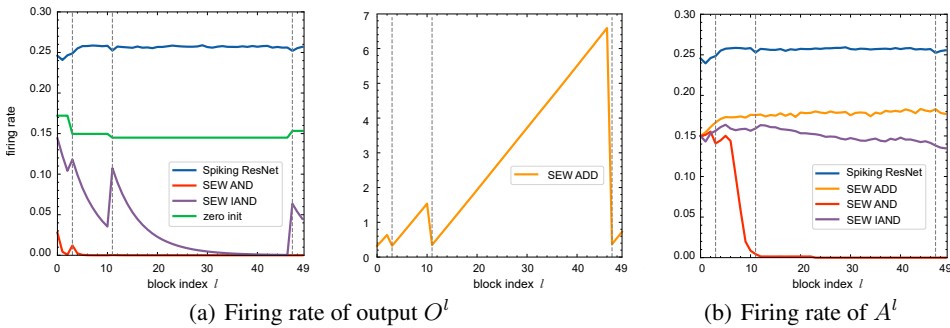

(a) Firing rate of output $O^l$                (b) Firing rate of $A^l$

Figure 5: The initial firing rates of output $O^l$ and $A^l$ in $l$-th block on 152-layer network.

The surrogate gradient function we used in all experiments is $\sigma(x) = \frac{1}{\pi}\arctan(\frac{\pi}{2}\alpha x) + \frac{1}{2}$, thus $\sigma'(x) = \frac{\alpha}{2(1+(\frac{\pi}{2}\alpha x)^2)}$. When $V_{th} = 1, \alpha = 2$, the gradient amplitude $\left\|\frac{\partial L}{\partial S^l}\right\|$ of each block is shown in Fig. 6. Note that $\alpha = 2, \sigma'(x) \le \sigma'(0) = \sigma'(1 - V_{th}) = 1$ and $\sigma'(0 - V_{th}) = 0.092 < 1$. It can be found that the gradients in Spiking ResNet-152 decay from deeper layers to shallower layers in the identity mapping areas without zero initialization, which is caused by $\sigma'(x) \le 1$. It is worth noting that the decay also happens in Spiking ResNet-152 with zero initialization. The small convex $\bigwedge$ near the dotted lines is caused by the vanishing gradients of those $S_j^l[t] = 0$. After these gradients decays to 0 completely, $\left\|\frac{\partial L}{\partial S^l}\right\|$ will be a constant because the rest gradients are calculated by $S_j^l[t] = 1$ and $\sigma'(1 - V_{th}) = 1$, which can also explain why the gradient-index curve is horizontal at some areas. When referring to SEW ResNet-152 with zero initialization, it can be found that all gradient-index curves are similar no matter what $g$ we choose. This is caused by that in the identity mapping areas, $S^l$ is constant for all index $l$, and the gradient also becomes a constant as it will not flow through SNs. Without zero initialization, the vanishing gradient happens in the SEW AND ResNet-152, which is caused by the silence problem. The gradients of SEW ADD, IAND network increase slowly when propagating from deeper layers to shallower layers, due to the adequate firing rates shown in Fig. 5.

When $V_{th} = 0.5, \alpha = 2$, $\sigma'(0 - V_{th}) = \sigma'(1 - V_{th}) = 0.288 < 1$, indicating that transmitting spikes to SNs is prone to causing vanishing gradient, as shown in Fig. 7. With zero initialization, the decay in Spiking ResNet-152 is more serious because gradient from $\mathcal{F}^l$ can not contribute. The SEW ResNet-152 will not be affected no matter what $g$ we choose. When $V_{th} = 1, \alpha = 3$, $\sigma'(1 - V_{th}) = 1.5 > 1$, indicating that transmitting spikes to SNs is prone to causing exploding gradient. Fig. 8 shows the gradient in this situation. Same with the reason in Fig. 6, the change of surrogate function will increase gradients of all networks without zero initialization, but not affect SEW ResNet-152 with zero initialization. The Spiking ResNet-152 meets exploding gradient, while this problem in SEW ADD, IAND ResNet-152 is not serious.

## 4.2 DVS Gesture Classification

The origin ResNet, which is designed for classifying the complex ImageNet dataset, is too large for the DVS Gesture dataset. Hence, we design a tiny network named 7B-Net, whose structure is *c32k3s1-BN-PLIF-{SEW Block-MPk2s2}*7-FC11*. Here c32k3s1 means the convolutional layer with channels 32, kernel size 3, stride 1. MPk2s2 is the max pooling with kernel size 2, stride 2. The symbol {}*7 denotes seven repeated structure, and PLIF denotes the Parametric Leaky-Integrate-and-Fire Spiking Neuron with a learnable membrane time constant, which is proposed in [8] and can be described by Eq. (5). See Sec.A.1 for AER data pre-processing details.

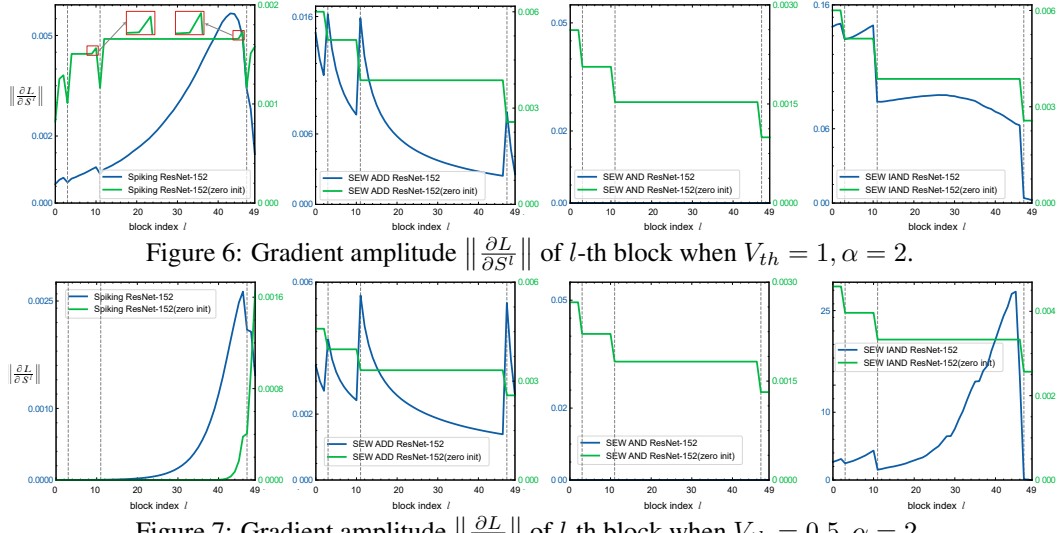

Figure 6: Gradient amplitude $\left\|\frac{\partial L}{\partial S^l}\right\|$ of $l$-th block when $V_{th} = 1, \alpha = 2$.

Figure 7: Gradient amplitude $\left\|\frac{\partial L}{\partial S^l}\right\|$ of $l$-th block when $V_{th} = 0.5, \alpha = 2$.

**Spiking ResNet *vs.* SEW ResNet.** We first compare the performance of SEW ResNet with *ADD* element-wise function (SEW ADD ResNet) and Spiking ResNet by replacing SEW blocks with basic blocks. As shown in Fig. 9 and Tab. 4, although the training loss of Spiking ResNet (blue curve) is lower than SEW ADD ResNet (orange curve), the test accuracy is lower than SEW ADD ResNet (90.97% v.s. 97.92%), which implies that Spiking ResNet is easier to overfit than SEW ADD ResNet.

**Evaluation of different element-wise functions and plain block.** As the training cost of SNNs on the DVS Gesture dataset is much lower than on ImageNet, we carry out more ablation experiments on the DVS Gesture dataset. We replace SEW blocks with the plain blocks (no shortcut connection) and test the performance. We also evaluate all kinds of element-wise functions $g$ in Tab. 1. Fig. 9 shows the training loss and training/test accuracy on DVS Gesture. The sharp fluctuation during early epochs is caused by the large learning rate (see Sec.A.1). We can find that the training loss is SEW IAND<Spiking ResNet<SEW ADD<Plain Net<SEW AND. Due to the overfitting problem, a lower loss does not guarantee a higher test accuracy.

Tab. 4 shows the test accuracy of all networks. The SEW ADD ResNet gets the highest accuracy than others.

**Comparisons with State-of-the-art Methods.** Tab. 5 compares our network with SOTA methods. It can be found that our SEW ResNet outperforms the SOTA works in accuracy, parameter numbers, and simulating time-steps.

| Network | Element-Wise Function $g$ | Accuracy(%) |
|---|---|---|
| SEW ResNet | ADD | 97.92 |
| SEW ResNet | IAND | 95.49 |
| Plain Net | – | 91.67 |
| Spiking ResNet | – | 90.97 |
| SEW ResNet | AND | 70.49 |

Table 4: Test accuracy on DVS Gesture. The networks' order is ranked by accuracy.

| Network | Accuracy(%) | Parameters | $T$ |
|---|---|---|---|
| **c32k3s1-BN-PLIF-{SEW Block (c32) -MPk2s2}*7-FC11 (7B-Net)** | 97.92 | 0.13M | 16 |
| {c128k3s1-BN-PLIF-MPk2s2}*5-DP-FC512-PLIF-DP-FC110-PLIF-APk10s10 [8] | 97.57 | 1.70M | 20 |
| Spiking ResNet-17 with td-BN [64] | 96.87 | 11.18M | 40 |
| MPk4-c64k3-LIF-c128k3-LIF-APk2-c128k3-LIF-APk2-FC256-LIF-FC11[16] | 93.40 | 23.23M | 60 |

Table 5: Comparison with the state-of-the-art (SOTA) methods on DVS Gesture dataset.

### 4.3 CIFAR10-DVS Classification

We also report SEW ResNet on the CIFAR10-DVS dataset, which is obtained by recording the moving images of the CIFAR-10 dataset on a LCD monitor by a DVS camera. As CIFAR10-DVS is more complicated than DVS Gesture, we use the network structure named Wide-7B-Net, which is similar to 7B-Net but with more channels. The structure of Wide-7B-Net is *c64k3s1-BN-PLIF-{SEW Block (c64)-MPk2s2}*4-c128k3s1-BN-PLIF-{SEW Block (c128)-MPk2s2}*3-FC10.*

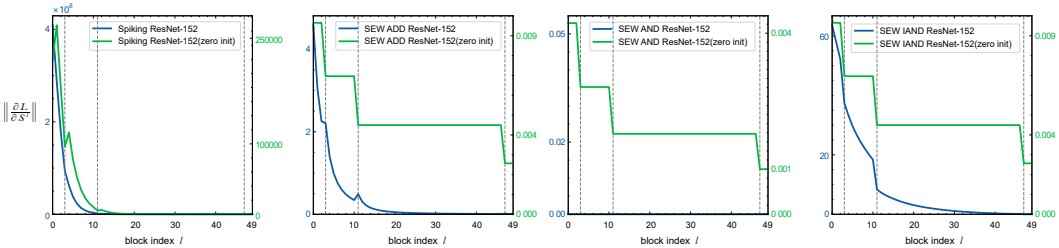

Figure 8: Gradient amplitude $\left\| \frac{\partial L}{\partial S^l} \right\|$ of $l$-th block when $V_{th} = 1, \alpha = 3$.

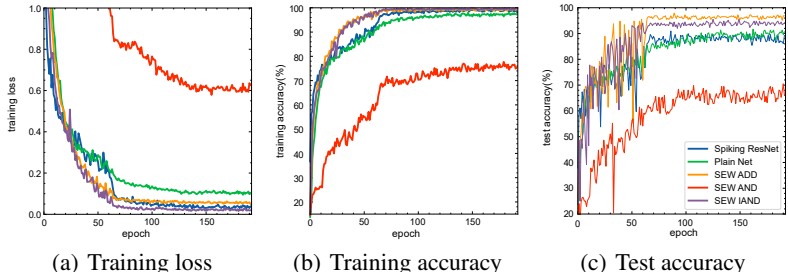

| (a) Training loss | (b) Training accuracy | (c) Test accuracy |
|---|---|---|

Figure 9: Comparison of the training loss, training accuracy and test accuracy on DVS Gesture dataset.

| Network | Accuracy(%) | Parameters | $T$ |
|---|---|---|---|
| **c64k3s1-BN-PLIF-{SEW Block (c64)-MPk2s2}*4-c128k3s1-BN-PLIF-{SEW Block (c128)-MPk2s2}*3-FC10 (Wide-7B-Net)** | 64.8, 70.2, 74.4 | 1.19M | 4, 8, 16 |
| {c128k3s1-BN-PLIF-MPk2s2}*4-DP-FC512-PLIF-DP-FC100-PLIF-APk10s10[8] | 74.8 | 17.4M | 20 |
| Spiking ResNet-19 with td-BN [64] | 67.8 | 11.18M | 10 |

Table 6: Comparison with the state-of-the-art (SOTA) methods on CIFAR10-DVS dataset.

In Tab.6, we compare SEW ResNet with the previous Spiking ResNet. One can find that our method achieves better performance (70.2% v.s. 67.8%) and fewer time-steps (8 v.s. 10) than the Spiking ResNet [64]. We also compare our method with the state-of-the-art (SOTA) supervised learning methods on CIFAR10-DVS. The accuracy of our Wide-7B-Net is slightly lower than the current SOTA method [8] (74.4% v.s. 74.8%), which uses 1.25 times as many simulation time-steps $T$ (20 v.s. 16) and 14.6 times as many the number of parameters (17.4M v.s. 1.19M). Moreover, when reducing $T$ shapely to $T = 4$, our Wide-7B-Net can still get the accuracy of 64.8%.

# 5  Conclusion

In this paper, we analyze the previous Spiking ResNet whose residual block mimics the standard block of ResNet, and find that it can hardly implement identity mapping and suffers from the problems of vanishing/exploding gradient. To solve these problems, we propose the SEW residual block and prove that it can implement the residual learning. The experiment results on ImageNet, DVS Gesture, and CIFAR10-DVS datasets show that our SEW residual block solves the degradation problem, and SEW ResNet can achieve higher accuracy by simply increasing the network's depth. Our work may shed light on the learning of "very deep" SNNs.

# 6  Acknowledgment

This work is supported by grants from the National Natural Science Foundation of China under contracts No.62027804, No.61825101, and No.62088102.

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
