# A Appendix

## A.1 Hyper-Parameters

For all datasets, the surrogate gradient function is $\sigma(x) = \frac{1}{\pi}\arctan(\frac{\pi}{2}\alpha x) + \frac{1}{2}$, thus $\sigma'(x) = \frac{\alpha}{2(1+(\frac{\pi}{2}\alpha x)^2)}$, where $\alpha$ is the slope parameter. We set $\alpha = 2$, $V_{reset} = 0$ and $V_{th} = 1$ for all neurons. The optimizer is SGD with momentum 0.9. As recommended by [62], we detach $S[t]$ in the neuronal reset Eq. (3) in the backward computational graph to improve performance. We use the mixed precision training [36], which will accelerate training and decrease memory consumption, but may cause slightly lower accuracy than using full precision training. The hyper-parameters of the SNNs for different datasets are shown in Tab. 7. Tab. 8 shows the learning rates of the SNNs with different element-wise functions for DVS Gesture. The data pre-processing methods for three datasets are as following:

**ImageNet** The data augmentation methods used in [14] are also applied in our experiments. A 224×224 crop is randomly sampled from an image or its horizontal flip with data normalization for train samples. A 224×224 resize and central crop with data normalization is applied for test samples.

**DVS128 Gesture** We use the same AER data pre-processing method as [8], and utilize *random temporal delete* to relieve overfitting, which is illustrated in Sec. A.2.

**CIFAR10-DVS** We use the same AER data pre-processing method as DVS128 Gesture. We do not use *random temporal delete* because CIFAR10-DVS is obtained by static images.

| Dataset | Learning Rate Scheduler | Epoch | $lr$ | Batch Size | $T$ | $n_{gpu}$ |
|---------|------------------------|-------|------|-----------|-----|-----------|
| ImageNet | Cosine Annealing [35], $T_{max} = 320$ | 320 | 0.1 | 32 | 4 | 8 |
| DVS Gesture | Step, $T_{step} = 64.\gamma = 0.1$ | 192 | 0.1 | 16 | 16 | 1 |
| CIFAR10-DVS | Cosine Annealing, $T_{max} = 64$ | 64 | 0.01 | 16 | 4, 8, 16 | 1 |

Table 7: Hyper-parameters of the SNNs for three datasets.

## A.2 Random Temporal Delete

To reduce overfitting, we propose a simple data augmentation method called *random temporal delete* for sequential data. Denote the sequence length as $T$, we randomly delete $T - T_{train}$ slices in the origin sequence and use $T_{train}$ slices during training. During inference we use the whole sequence, that is, $T_{test} = T$. We set $T_{train} = 12, T = 16$ in all experiments on DVS Gesture.

Fig. 10 compares the training loss and training/test accuracy of Plain Net, Spiking ResNet, and SEW ResNet with or without *random temporal delete* (RTD). Here the element-wise function $g$ is *ADD*. It can be found that the network with RTD has higher training loss and lower training accuracy than the network without RTD, because RTD can increase the difficulty of training. The test accuracy of the network with RTD is higher than that without RTD, showing that RTD will reduce overfitting. The results on the three networks are consistent, indicating that RTD is a general sequential data augmentation method.

## A.3 Firing rates on DVS Gesture

Fig. 11(a) shows the firing rates of $A^l$ in each block from 7B-Net for DVS Gesture. Note that if $g$ is *AND*, the SEW block gets closer to identity mapping when the firing rate approaches 1, while for other

| Network | Element-Wise Function $g$ | Learning Rate |
|---------|--------------------------|---------------|
| SEW ResNet | ADD | 0.001 |
| SEW ResNet | AND | 0.03 |
| SEW ResNet | IAND | 0.063 |
| Spiking ResNet | - | 0.1 |
| Plain Net | - | 0.005 |

Table 8: Learning rates of the SNNs for DVS Gesture.

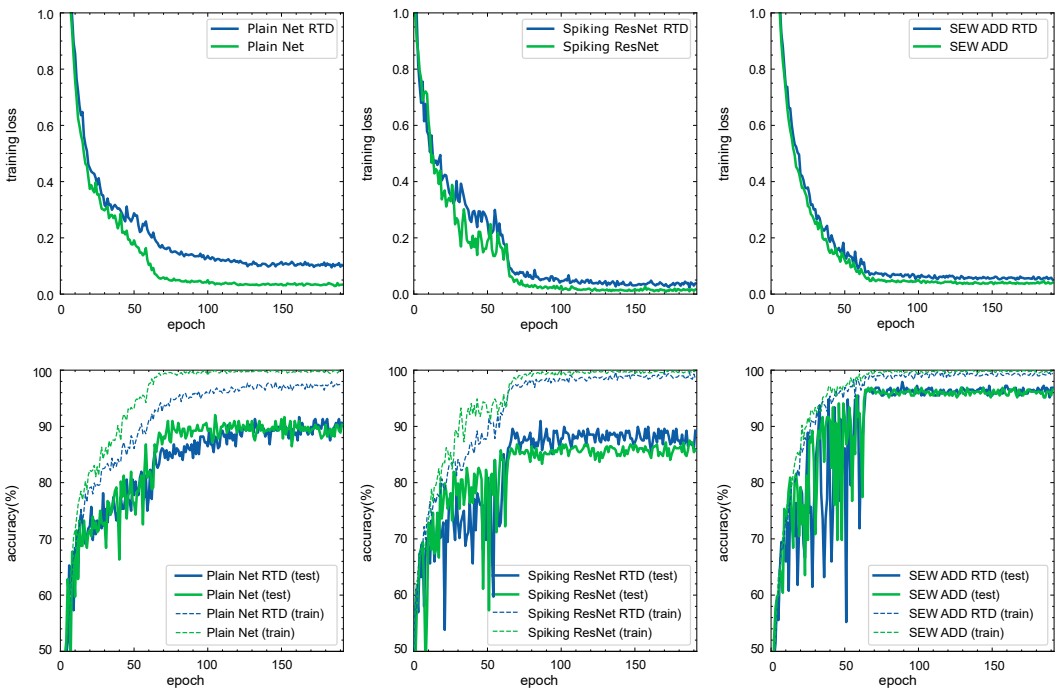

Figure 10: Comparison of training loss and training/test accuracy with/without random temporal delete (RTD).

$g$, the SEW block becomes identity mapping when the firing rate approaches 0. When all SEW blocks become identity mapping, the 7B-Net will become *c32k3s1-BN-PLIF-{MPk2s2}\*7-FC11*, which is a too simple network to cause underfitting. Thus, the SEW blocks in 7B-Net are not necessary to be identity mapping. Fig. 11(b) shows the firing rates of each block's output $O^l$. The firing rates do not strictly decrease with block index increases as blocks are connected by max pooling, which squeezes sparse spikes and increases the firing rate. It can be found that the blocks in SEW AND network have the lowest firing rates. The blocks in SEW IAND network have higher firing rates than those of SEW AND network, and the SEW IAND network has much higher accuracy than the SEW AND network (95.49% v.s. 70.49%), indicating that using *IAND* to replace *AND* can relieve the silence problem discussed in Sec.4.1.

## A.4 Gradients in Spiking ResNet with Firing Rates

The gradients of SNNs are affected by firing rates, which is the reason why we analyze the firing rates before gradients in Sec.4.1. Consider a spiking ResNet with $k$ sequential blocks to transmit $S^l[t]$, and the identity mapping condition is met, e.g., the spiking neurons are the IF neurons with $0 < V_{th} \leq 1$, then we have $S^l[t] = S^{l+1}[t] = ... = S^{l+k-1}[t] = O^{l+k-1}[t]$. We get

$$\frac{\partial O_j^l[t]}{\partial S_j^l[t]} = \frac{\partial \text{SN}(S_j^l[t])}{\partial S_j^l[t]} = \Theta'(S_j^l[t] - V_{th}) \tag{12}$$

$$\frac{\partial L}{\partial S_j^l[t]} = \frac{\partial L}{\partial O_j^l[t]}\Theta'(S_j^l[t] - V_{th}). \tag{13}$$

Then the gradients between two adjacent blocks are

$$\frac{\partial L}{\partial O^{l+i}} = \frac{\partial L}{\partial O^{l+i+1}}\Theta'(S^{l+i+1} - V_{th}). \tag{14}$$

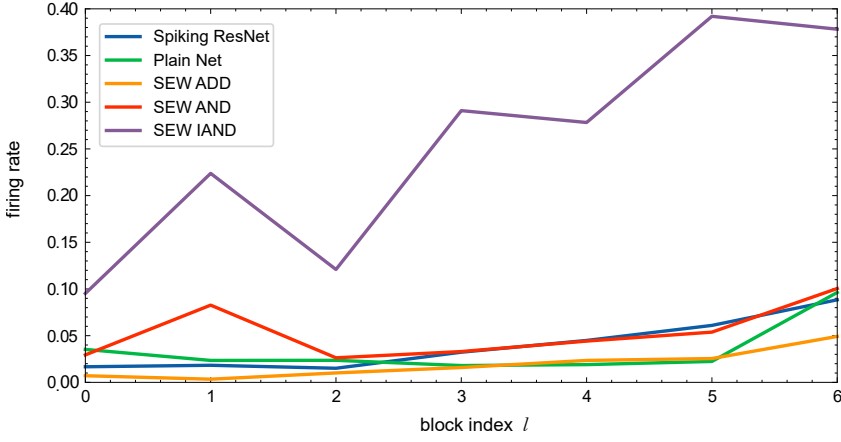

(a) Firing rates of $A^l$ in each block on DVS Gesture Gesture

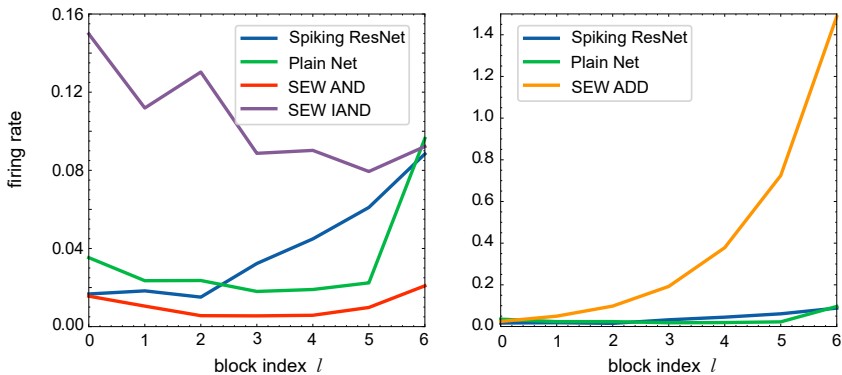

(b) Firing rates of the output $O^l$ in each block on DVS Gesture Gesture

Figure 11: Firing rates of the last SN and the output $O^l$ in each block of 7B-Net on DVS Gesture.

Denote the number of neurons as $N$, the firing rate of $S^l$ as $\Phi = \frac{\sum_{j=0}^{N-1} \sum_{t=0}^{T-1} S_j^l[t]}{NT}$, then

$$\left\| \frac{\partial L}{\partial S^l} \right\| = \left\| \frac{\partial L}{\partial O^{l+k-1}} \right\| \cdot \left\| \prod_{i=0}^{k-1} \Theta'(S^{l+i} - V_{th}) \right\|, \tag{15}$$

where

$$\left\| \prod_{i=0}^{k-1} \Theta'(S^{l+i} - V_{th}) \right\| = \sqrt{NT\Phi(\Theta'(1 - V_{th}))^{2k} + NT(1 - \Phi)(\Theta'(0 - V_{th}))^{2k}}$$

$$\rightarrow \begin{cases} \sqrt{NT}, & \Theta'(1 - V_{th}) = 1, \Theta'(0 - V_{th}) = 1 \\ \sqrt{NT\Phi}, & \Theta'(1 - V_{th}) = 1, \Theta'(0 - V_{th}) < 1 \\ \sqrt{NT(1 - \Phi)}, & \Theta'(1 - V_{th}) < 1, \Theta'(0 - V_{th}) = 1 \\ 0, & \Theta'(1 - V_{th}) < 1, \Theta'(0 - V_{th}) < 1 \\ +\infty, & \Theta'(1 - V_{th}) > 1 \ or \ \Theta'(0 - V_{th}) > 1. \end{cases}$$

### A.5  0/1 Gradients Experiments

As the analysis in Sec.3.2 shows, the vanishing/exploding gradient problems are easy to happen in Spiking ResNet because of accumulative multiplication. A potential solution is to set $\Theta'(0 - V_{th}) = \Theta'(1 - V_{th}) = 1$. Specifically, we have trained the Spiking ResNet on ImageNet by setting $V_{th} = 0.5$ and $\sigma'(x) = \frac{1 + \frac{\pi^2}{4}}{1 + (\pi x)^2}$ in the last SN of each block to make sure that $\Theta'(0 - V_{th}) = \Theta'(1 - V_{th}) = 1$.

| Surrogate function | SEW ResNet (ADD) | Spiking ResNet |
|---|---|---|
| ArcTan | 0.8263 | 0.7733 |
| Rectangular | 0.8256 | 0.6601 |
| Constant 1 | 0.1256 | 0.1 |

Table 9: Test accuracy of SEW ADD ResNet and Spiking ResNet on CIFAR-10 with different surrogate functions.

However, this network will not converge, which may be caused by that SNNs are sensitive to surrogate functions.

[64] uses the Rectangular surrogate function $\sigma'(x) = \frac{1}{a}\text{sign}(|x| < \frac{a}{2})$. If we set $a = 1$, then $\sigma'(x) \in \{0, 1\}$. According to Eq.(8), using this surrogate function can avoid the gradient exploding/vanishing problems in Spiking ResNet. We compare different surrogate functions, including Rectangular ($\sigma'(x) = \text{sign}(|x| < \frac{1}{2})$), ArcTan ($\sigma'(x) = \frac{1}{1+(\pi x)^2}$) and Constant 1 ($\sigma'(x) \equiv 1$), in the SNNs on CIFAR-10. Note that we aim to evaluate 0/1 gradients, rather than achieve SOTA accuracy. Hence, we use a lightweight network, whose structure is *c32k3s1-BN-IF-{{SEW Block (c32)}*2-MPk2s2}*5-FC10*. We use *ADD* as $g$ in SEW blocks. We also compare with Spiking ResNet by replacing SEW blocks with basic blocks. The results are shown in Tab.9. The learning rates for each surrogate function are fine-tuned.

Tab.9 indicates that the choice of surrogate function has a considerable influence on the SNN's performance. Although Rectangular and Constant 1 can avoid the gradient exploding/vanishing problems in Eq.(8), they still cause lower accuracy or even make the optimization not converges. Tab.9 also shows that the SEW ResNet is more robust to the surrogate gradient as it always has higher accuracy than the Spiking ResNet with the same surrogate function.

## A.6 Reproducibility

All experiments are implemented with SpikingJelly [7], which is an open-source deep learning framework for SNNs based on PyTorch [41]. Source codes are available at `https://github.com/fangwei123456/Spike-Element-Wise-ResNet`. To maximize reproducibility, we use identical seeds in all codes.