# OpenReview forum: "Deep Residual Learning in Spiking Neural Networks"
_NeurIPS.cc/2021/Conference — NeurIPS 2021 Poster_

### Official Review · Reviewer_uG22 · 2021-07-06

**Rating:** 7
**Confidence:** 2

**Summary:**

The present paper is concerned about a residual architecture for spiking neural networks. First, the authors point out two drawbacks of the existing Spiking ResNet. The first drawback is that Spiking ResNet cannot easily represent identity mapping for some neuronal dynamics. The second drawback is that even if a block of Spiking ResNet represents the identity mapping, the gradient of its output with respect to the input is not equal to 1, and therefore, a sequence of Spiking ResNet blocks suffers from vanishing or exploding gradients. These two drawbacks hinder us from training a deep Spiking ResNet.
The authors present the spike-element-wise (SEW) residual block to address the drawbacks mentioned above. Since the drawbacks are caused by the architecture where the skip connection is fed into a spiking neuron, the idea is to skip the spiking neuron and directly add/multiply/IAND the residual mapping and the input spike sequence. The authors discussed that the SEW block can address the drawbacks.
The authors present two sets of experiments, one uses ImageNet and the other uses DVS Gesture data set. The experimental results suggest that the proposed SEW ResNet successfully addresses the drawbacks and the deeper the network is the better the performance is.


**Main Review:**

I cannot judge whether the proposed architecture is novel or not, because I am working on a learning algorithm of SNN rather than its architecture. However, provided that the proposed architecture is novel, I think this paper is good because it clearly states the drawbacks of the existing method and the proposed method successfully addresses it.

One minor concern is that there are many typos;
- l.271 "indicating that indicating that" -> "indicating that"
- l.275 "Fig. 7 shows" -> "Fig. 8 shows" ?
- l.276 "Fig. 8" -> "Fig. 6" ?
to name a few. I would recommend the authors to go through the paper to improve the quality of the paper.

**Time Spent Reviewing:**

4 hours

---

> ### Author Response · Authors · 2021-08-09
> **Response to Reviewer uG22**
>
> We appreciate your constructive comments. We would like to address your concerns below.
>
> > However, provided that the proposed architecture is novel, I think this paper is good because it clearly states the drawbacks of the existing method and the proposed method successfully addresses it.
>
> We believe the proposed architecture is novel. As acknowledged by Reviewer 96mQ, SEW structure is the core contribution in this paper. The existing works of Spiking ResNet mimic the residual block in ANNs and replace ReLU activation layers with spiking neurons, which suffers the problem of vanishing/exploding gradient. We propose the SEW ResNet to address this problem.
>
> > I would recommend the authors to go through the paper to improve the quality of the paper.
>
> Thanks, we have gone through the paper and revised them.

---

> > ### Comment · Reviewer_uG22 · 2021-08-09
> > **Response to the rebuttal**
> >
> > I appreciate the authors to clarify my concerns, and I will keep my score. Below, I would like to elaborate on four dimensions specified in the reviewer guideline for clarity (note: there is no additional requirement for the authors).
> >
> > # Originality
> > The main contribution of this paper is the SEW residual block, which addresses the two drawbacks of the existing spiking ResNet. As far as I am aware of, this structure is novel.
> >
> > # Quality
> > Both the drawbacks of the existing architecture and the ideas to address them are explained theoretically, which is easy to understand for me and looks sound. Furthermore, the authors conduct comprehensive experiments to validate these ideas, and in specific, the authors succeed to train deeper SNNs, which is convincing to me.
> >
> > # Clarity
> > As I mentioned in the initial review, the paper had many typos. However, the structure itself is solid and it is easy to understand the logic.
> >
> > # Significance
> > Although the idea looks simple (note: I like simple ideas), this paper reasonably addresses drawbacks of the existing ResNet-like architecture for SNNs and enables us to train deeper SNNs. Therefore, I believe it is significant to the SNN community.

---

### Official Review · Reviewer_96mQ · 2021-07-06

**Rating:** 8
**Confidence:** 5

**Summary:**

This paper wants to apply the residual block into deep spiking neural networks to achieve better classification accuracy. They proposed a spike element-wise (SEW) to realize residual learning. This top and idea are very important in training a robust deep spiking neural network and the residual block is approved useful in the deep learning field. The authors evaluated the SEW on several datasets such as ImageNet and DVS gesture datasets, the experimental results show the proposed method can achieve better performance.


**Limitations And Societal Impact:**

This paper still existed some problems that I hope the authors could illustrate in a clearer way.

1.	The authors argued that they were the first time to directly training deep SNNs with more than 100 layers. I don’t think this is the core contribution in this paper, because of the residual block, the spiking could be deeper. In my opinion, SEW structure is the most important point in this paper, and directly training a 50-layer and 100-layer snn is not a huge breakthrough. Otherwise, if they could give a more detailed analysis about why other methods can’t train a 100-layer snn except section 3.2, it may be more reasonable.
2.	Why the RBA block can be seen as a special case of the SEW block? I mean SEW is another kind of RBA with binary input and output.
3.	Equ. 11 is wonderful, how about other bit operations?
4.	Fig. 5 a seems strange, please give more explanations.
5.	When the input is aer format, how did you deal with DVS input?
6.	If you can analyze the energy consumption as reference[15] did, this paper would be more solid.


**Main Review:**

A spiking neural network (SNN) is considered a promising approach to build an intelligent machine as efficient as the human brain. This paper investigated the learning of deep SNN by converting residual network which allowed a network to go extremely deep and has achieved great success in pattern classification tasks. This paper analyzed the drawbacks of some current spiking resnet methods, aiming at this, the paper introduced SEW block as formulated as equation 9. Through the identity mapping with ADD, AND, and IAND, the binary spiking signals could be calculated and communicated between neural layers.

Different from ANN2SNN methods, this paper applied a spiking bp-based method to training the proposed network model. Table 3 shows the experimental results on ImageNet, the authors reported impressive results that the classification accuracy is positive with fewer time steps.


**Time Spent Reviewing:**

1

---

> ### Author Response · Authors · 2021-08-09
> **Response to Reviewer 96mQ**
>
> Thank you for your positive and thoughtful comments. We are delighted that you find our idea very important and the results great success and impressive. We would like to address your concerns and answer your questions in the following.
>
> > In my opinion, SEW structure is the most important point in this paper, and directly training a 50-layer and 100-layer snn is not a huge breakthrough. Otherwise, if they could give a more detailed analysis about why other methods can’t train a 100-layer snn except section 3.2, it may be more reasonable.
>
> Thanks, we agree and appreciate that the reviewer regards the SEW structure as the core contribution of our paper. As illustrated in Sec.3.2 and lines 234-279, we believe the difficulty of training deep spiking ResNet is the gradient vanishing/exploding problem, which can be overcome by the proposed SEW ResNet. We have added the experiments to evaluate the performance of the Spiking ResNet-101/152 and SEW ResNet 152 (see Tab.R3 and To All Reviewers). The results show that the accuracy of deeper Spiking ResNet decays seriously. Specifically, the test accuracy of Spiking ResNet-101 and Spiking ResNet-152 on Imagenet are only 31.79% and 10.03%, indicating that training deep Spiking ResNet is almost impossible. Hence, we regard the directly trained SNN with more than 100 layers as a contribution.
>
> > Why the RBA block can be seen as a special case of the SEW block? I mean SEW is another kind of RBA with binary input and output.
>
> The RBA block can be obtained by replacing SN in the SEW block with the ReLU activation layer and setting ADD as $g$. We agree with the reviewer that the SEW block also can be seen as another kind of RBA block.  We clarified this in Section 3.3 and changed it to "the SEW block is an extension of the RBA block".
>
> > Equ. 11 is wonderful, how about other bit operations?
>
> Thanks. We have considered more bit operations, including OR, XOR, and MAX. However, the gradient of OR and XOR is undefined as they can not be expressed by the four fundamental operations of arithmetic. The MAX operation is widely used in pooling, and its gradient is defined by:
>
> $$
> \begin{align}
> 	\frac{\partial \mathrm{MAX}(X, Y)}{\partial X} =
> 	\begin{cases}
> 		1, \text{if} X \geq Y \\\\
> 		0, \text{if} X<Y
> 	\end{cases}.
> \end{align}
> $$
>
> When the identity mapping condition is met, the gradient of the output of the $(l+k-1)$-th SEW block with respect to the input of the $l$-th SEW block can be calculated layer by layer:
> $$
> \begin{align}
> 	\frac{\partial O_{j}^{l+k-1}[t]}{\partial S_{j}^{l}[t]}  = \prod_{i=0}^{k-1}\frac{\partial \mathrm{MAX}(A_{j}^{l+i}[t], S_{j}^{l+i}[t])}{\partial S_{j}^{l+i}[t]} =
> 	\prod_{i=0}^{k-1}\frac{\partial \mathrm{MAX}(0, S_{j}^{l+i}[t])}{\partial S_{j}^{l+i}[t]}
> 	= 1.
> \end{align}
> $$
> It can be found that the SEW ResNet with MAX as $g$ can also overcome the vanishing/exploding gradient problem. However, using MAX will cause too fast saturation as $O^{l}[t] =\mathrm{MAX} \left ( {\rm SN}(\mathcal{F}(O^{l-1}[t])), O^{l-1}[t] \right ) \geq O^{l-1}[t]$. Consequently, it will make the firing rate of SNs in deep layers reach 1 quickly and decrease the accuracy.
>
> > Fig. 5a seems strange, please give more explanations.
>
> To verify that SEW ResNet can overcome vanishing/exploding gradient, we firstly analyze the firing rates. Fig.5(a) illustrates the firing rate of output $O^{l}$ in $l$-th block on Spiking ResNet-152 and SEW ResNet-152. When using zero initialization, the forward propagation of the basic block and SEW block (see Fig.1) are identical. Thus, Spiking ResNet, SEW AND ResNet, SEW IAND ResNet, and SEW ADD ResNet have the same firing rates (the green curve in Fig.5(a) left). However, without zero initialization, their firing rates are different (the blue, red, and purple curves in Fig.5(a) left and the yellow curve in Fig.5(a) right). Note that the output $O^{l}$ in  $l$-th block on SEW ADD ResNet (the yellow curve in Fig.5(a) right) is spikes accumulation rather than a spike. Nevertheless, we can still calculate the "firing rate" of $O^{l}$ by averaging on time-steps and neuron numbers. We clarify this in Section 4.1.
>
> > When the input is AER format, how did you deal with DVS input?
>
> We use the same AER data processing method in Ref.[6]. Specifically, we use $T$ time steps and build $T$ frames containing the event count for each time step, position, and polarity. The compared methods in Table 5 also use similar event-to-frame integration methods.
>
> > If you can analyze the energy consumption as reference[15] did, this paper would be more solid.
>
> Thanks for your suggestion. We have added the analysis of energy consumption.  According to Ref.[15], we utilize 12.5pJ/FLOP for ANN and 77fJ/SOP for SNN as the power consumption baseline. The following table compares the energy consumption of ResNet and SEW ADD ResNet. Note that we do not consider the memory access energy in our study because it is dependent on the hardware. Yet we are aware that SNNs incur significant data movement because the membrane potentials need to be fetched at every time-step.
>
> | ResNet Structure | 18       | 34       | 50       | 101      | 152      |
> | ---------------- | -------- | -------- | -------- | -------- | -------- |
> | ANN OP (GFLOP)   | 1.82     | 3.68     | 4.14     | 7.87     | 11.61    |
> | SNN OP (GSOP)    | 1.41     | 3.88     | 4.83     | 9.3      | 13.72    |
> | ANN Power (mJ)   | 22.75    | 46       | 51.75    | 98.375   | 145.125  |
> | SNN Power (mJ)   | 0.10857  | 0.29876  | 0.37191  | 0.7161   | 1.05644  |
> | A/S Power Ratio  | 209.5422 | 153.9697 | 139.1466 | 137.3761 | 137.3717 |
>
> **Table R5: Comparison of power consumption.**
>
> The results show that although using ADD will make $O^{l}$ have more non-zero elements, the SEW ADD ResNet still has a significant advantage on energy consumption (Tab.R5). Compared with the ANN2SNN method (T=350) that has 9 times energy efficiency than ANN (Ref. [15]), our SEW ADD ResNet (T=4) can reach 137-210 times energy efficiency than ANN.

---

> > ### Comment · Reviewer_96mQ · 2021-08-10
> > **Response to the rebuttal**
> >
> > I appreciate the rebuttal response by the authors. The authors addressed my concerns very well, especially the analysis of the power consumption. From table R5 in their response, we can see that the proposed spiking residual model consumes lower GSOP and energy according to my suggestion.
> >
> > All in all, this paper proposed a very important idea to combine the residual block into spiking-based models. From the experimental results, the proposed structure becomes very effective in practice, and the authors addressed my concerns in a very clear way. I would like to raise my score from 6 to 8, and willing to see this paper could be accepted by NeurIPS.

---

### Official Review · Reviewer_Up2E · 2021-07-13

**Rating:** 7
**Confidence:** 5

**Summary:**

In this paper, the authors proposed the spike-element-wise (SEW) ResNet to train deep SNNs with ResNet modules.  They achieved very excellent results compared to the SOTA. Although simple, the proposed methods turned out to be very effective in the practice.

**Limitations And Societal Impact:**

Not included. The authors may add a paragraph to discuss the limitation and potential negative societal impact of their work.

**Main Review:**

1. The methods and results were written clearly except for some minor grammar issues. Please check more carefully in the later version.

2. As the authors also state in lines 166-176, the SEW block is an extension of the RBA block. Thus, it suffers the same shortcoming that it leads to a non-negative output from the transform $\mathcal{F}$, which would impact the representational ability.

3. For the comparison with other methods, the simulation for SEW is with T=4. Is there any reason for not using T=6, the same as Spiking ResNet-34? Does the proposed method require more memory for training? Another issue here is that the references to the ANN2SNN methods are not accurate here. For example, Deng & Gu, 2021 ICLR can get 67.54% with T=128 and Li, 2021 ICML can get 71.12\% with T = 64.

4. The demonstration of gradient amplitude is a bit misleading. I suggest that the authors subtract the gradient calculated by $S_{j}^l[t]=1$ and $\sigma'(1-V_{th})=1$ out so that $0$ indicates vanishment.

5. The problem of SpikeNet may be relieved by choosing surrogate gradients for the identity map. This setup with the same training strategy as the SEW-Net can serve as a benchmark to compare as well.

6. It would be more informative to see further results on DVS-CIFAR-10.

**Time Spent Reviewing:**

10

---

> ### Author Response · Authors · 2021-08-09
> **Response to Reviewer Up2E**
>
> Thank you for your insightful and very detailed feedback. We are encouraged that you find our method very effective in the practice and the results excellent compared to the SOTA. We would like to address your concerns and answer your questions in the following.
>
> > Some minor grammar issues.
>
> Thanks for pointing this. We have gone through the paper and revised them.
>
> > The SEW block suffers the same shortcoming that it leads to a non-negative output from the transform $\mathcal{F}$, which would impact the representational ability.
>
> We agree that the RBA block suffers the non-decreasing problem, which is caused by the non-negative outputs of ReLU. Although the SEW block is an extension of the RBA block, one key difference between SN in the SEW block and ReLU in the RBA block is that the outputs of SN are no larger than 1. Consequently, as claimed in lines 172-176, this non-decreasing problem will never happen in SEW AND ResNet and SEW IAND ResNet. For SEW ADD ResNet, this problem can be relieved as the outputs of $k$ sequential SEW blocks are no larger than $k$. Besides, a downsample SEW block will regulate the outputs to be no larger than 2.
>
> > Is there any reason for not using T=6, the same as Spiking ResNet-34?
>
> The main reason we use a smaller T is to illustrate the superiority of our method. As most supervised learning methods for SNNs are based on
> BPTT (backpropagation through time), the memory and time complexity are proportional to $T$. Thus, the smaller $T$ is desirable in SNNs but may decrease the network’s capacity. However, we show in Table 3 that our network can achieve higher accuracy than Spiking ResNet in Ref.[56], even with smaller $T$.
>
> In our experiment, we use layer-by-layer propagation to train SEW ResNet, which is provided by SpikingJelly (see Ref.[1] in the appendix for more details). The actual batch size is $TN$ for stateless layers (e.g., convolutional layers), where $T$ is the total time-steps and $N$ is the batch size. For the sake of computing efficiency on GPU, $TN$ should be a power of 2. Considering that $N$ is usually set to a power of 2 in both ANNs and SNNs, we prefer to set $T$ to the power of 2 in all experiments. Thus we choose $T=4$ rather than $T=6$.
>
> > Does the proposed method require more memory for training?
>
> The proposed method does not require more memory for training. As illustrated in Fig.1, Eq.(7), and Eq.(9), the operations of SEW block are almost the same as those of the basic block in Spiking ResNet, but in different calculations orders. For example, when choosing ADD as $g$,  the operations of SEW block and the basic block are $SN(A^{l}[t]) + S^{l}[t]$ and $\mathrm{SN} (A^l[t] + S^{l}[t])$, respectively. These two kinds of computation graphs consume the same memory during training.  The downsample SEW block requires a little more memory than downsample basic block as it requires an extra SN (see Fig.2). However, the memory cost is negligible as there are only three downsample blocks in ResNet structure.
>
> > Another issue here is that the references to the ANN2SNN methods are not accurate here.
>
> Thanks for pointing to relevant works. We have added these two papers:
>
> [Deng and Gu, 2021] Shikuang Deng, and Shi Gu. Optimal conversion of conventional artificial neural networks to spiking neural networks. In International Conference on Learning Representations (ICLR), 2021.
>
> [Li et al., 2021] Yuhang Li, Shikuang Deng, Xin Dong, Ruihao Gong, and Shi Gu. A free lunch from ANN: Towards efficient, accurate spiking neural networks calibration. In International Conference on Machine Learning (ICML), 2021.
>
> > I suggest that the authors subtract the gradient calculated by $S^{l}_{j}[t]=1$ and $\sigma'(1 - V_{th})=1$ out so that 0 indicates vanishment.
>
> Thanks for your suggestion. We clarified this in our description.
>
> > The problem of SpikeNet may be relieved by choosing surrogate gradients for the identity map. This setup with the same training strategy as the SEW-Net can serve as a benchmark to compare as well.
>
> We agree. If we choose a specific surrogate function, e.g., $\Theta'(x) \equiv 1$, the gradient vanishing/exploding problem in Eq.(8) will be relieved. We have tested this method but found that Spiking ResNet with the surrogate function can not converge, caused by SNNs being sensitive to the surrogate function. Most popular surrogate functions (e.g., Sigmoid and SuperSpike) are an approximation of the Heaviside function, and their gradients have the bell shape (see Fig.3 in Ref.[34]). In contrast, the primitive function of $\Theta'(x) \equiv 1$ is a linear function, which is greatly different from the Heaviside function.
>
> > It would be more informative to see further results on DVS-CIFAR-10.
>
> Thanks. We have added the experiment on CIFAR10-DVS. Please refers to the section of To All Reviewers.

---

> > ### Comment · Reviewer_Up2E · 2021-08-12
> > **A few more questions**
> >
> > Thanks for the response to my questions. I have a few more questions based on the responses.
> >
> > >1. Is there any reason for not using T=6, the same as Spiking ResNet-34?
> >
> > I agree that T = 4 is enough to beat the other methods. I am just curious here that whether improving T would further improve the performance here. If it does, it can help demonstrate the flexibility of the proposed method with different T's.
> >
> > >2. The problem of SpikeNet may be relieved by choosing surrogate gradients for the identity map. This setup with the same training strategy as the SEW-Net can serve as a benchmark to compare as well.
> >
> > I am still a bit confused about why $\Theta'(x)\equiv 1$ causes the gradient vanishing/exploding problem. Considering the fact that other methods like TdBN are able to train an SNN, why the addition of $1$ to the surrogate gradient would cause such instability to the training procedure?

---

> > > ### Author Response · Authors · 2021-08-20
> > > **[2] Response to Reviewer To Up2E**
> > >
> > > > I am just curious here that whether improving T would further improve the performance here. If it does, it can help demonstrate the flexibility of the proposed method with different T's.
> > >
> > > Thanks for your suggestion. Because of the training cost, we did not test different $T$ on ImageNet. Instead, we have tested different $T$ on the CIFAR-10 dataset.  We aim to explore the effect of $T$ instead of achieving SOTA accuracy. Hence, we use a lightweight network, whose structure is *c32k3s1-BN-IF-{{SEW Block(c32k3s1)}\*2-MPk2s2}\*5-FC10*. The accuracy changes with respect to different $T$ ($1 \leq T \leq 32$) is shown in Tab.R8.
> > >
> > > | $T$  | 1     | 2      | 3      | 4      | 5      | 6     | 7      | 8     |
> > > | ---- | ----- | ------ | ------ | ------ | ------ | ----- | ------ | ----- |
> > > | Acc  | 0.737 | 0.7939 | 0.8133 | 0.8263 | 0.8334 | 0.846 | 0.8535 | 0.857 |
> > >
> > > | $T$  | 9      | 10     | 11     | 12     | 13     | 14    | 15         | 16     |
> > > | ---- | ------ | ------ | ------ | ------ | ------ | ----- | ---------- | ------ |
> > > | Acc  | 0.8598 | 0.8615 | 0.8634 | 0.8618 | 0.8627 | 0.861 | **0.8666** | 0.8634 |
> > >
> > > | $T$  | 17     | 18     | 19    | 20     | 21     | 22     | 23     | 24     |
> > > | ---- | ------ | ------ | ----- | ------ | ------ | ------ | ------ | ------ |
> > > | Acc  | 0.8663 | 0.8615 | 0.862 | 0.8587 | 0.8649 | 0.8566 | 0.8579 | 0.8587 |
> > >
> > > | $T$  | 25     | 26     | 27     | 28     | 29     | 30     | 31   | 32     |
> > > | ---- | ------ | ------ | ------ | ------ | ------ | ------ | ---- | ------ |
> > > | Acc  | 0.8594 | 0.8532 | 0.8589 | 0.8585 | 0.8559 | 0.8452 | 0.85 | 0.8473 |
> > >
> > > **Table R8: Test accuracy of SEW ADD ResNet on CIFAR-10 with different $T$.**
> > >
> > > It can be found that the accuracy firstly increases and then decreases slowly with the increase of $T$. This result of SEW ResNet is consistent with our previous experimental results on SNNs. A larger $T$ can increase the SNN's fitting ability, but too large $T$ does not guarantee better performance. We think the reasons are as follows:
> > >
> > > 1) A larger $T$ may cause over-fitting. 2) Gradients are prone to vanish (the long-term dependency problem of RNNs).
> > >
> > > > Considering the fact that other methods like TdBN are able to train an SNN, why the addition of 1 to the surrogate gradient would cause such instability to the training procedure?
> > >
> > > Ref.[62] uses the *Rectangular* surrogate function $\sigma'(x)=\frac{1}{a}sign(|x|<\frac{a}{2})$. If we set $a=1$, then $\sigma'(x) \in \{0,1\}$. According to Eq.(8), using this surrogate function can avoid the gradient exploding/vanishing problem in Spiking ResNet.  We also compare different surrogate functions, including *Rectangular* ($\sigma'(x)=sign(|x|<\frac{1}{2})$), *ArcTan* ($\sigma'(x)=\frac{1}{1 + (\pi x)^2})$ and *Constant 1* ($\sigma' (x)  \equiv 1$), in the SNN for CIFAR-10, which are shown in Tab.R9. Note that the learning rates for each surrogate function are fine-tuned.
> > >
> > > |             | SEW ResNet | Spiking ResNet |
> > > | ----------- | ---------- | -------------- |
> > > | ArcTan      | 0.8263     | 0.7733         |
> > > | Rectangular | 0.8256     | 0.6601         |
> > > | Constant 1  | 0.1256     | 0.1            |
> > >
> > > **Table R9: Test accuracy of SEW ADD ResNet and Spiking ResNet on CIFAR-10 with different surrogate functions.**
> > >
> > > Tab.R9 indicates that the choice of surrogate function has a considerable influence on the SNN's performance. Although *Rectangular* and *Constant 1* can avoid the gradient exploding/vanishing problem in Eq.(8), they still cause lower accuracy or even make the optimization not converges. Tab.R9 also shows that the SEW ResNet is more robust to the surrogate gradient as it always has higher accuracy than the Spiking ResNet with the same surrogate function.
> > >
> > > For the moment, the theory of surrogate learning has not been established, and there are only some experimental conclusions (e.g., Ref.[60] and [Wu et al., 2018]). We have tested the most frequently used surrogate functions provided by Ref.[1] of appendix in our early experiments, and chosen *ArcTan* for its better performance.
> > >
> > > [Wu et al., 2018] Yujie, Wu, et al. Spatio-temporal backpropagation for training high-performance spiking neural networks. Frontiers in neuroscience. 2018, 12: 331.

---

### Official Review · Reviewer_2Q6H · 2021-07-15

**Rating:** 6
**Confidence:** 3

**Summary:**

This work proposes spike-element-wise ResNet, in an effort to achieve identity mapping. The original Spiking ResNet put the element-wise operation before the activation function. However, with the LIF activation function, the activation will be binarized to spikes and thus the identity mapping is not fulfilled. Thus, the authors put the element-wise operation after the LIF activation.



**Limitations And Societal Impact:**

My concerns are listed above. In my opinion, it would be natural that multi-bit spikes outperform binary spikes, therefore I would reject this submission for now. If the authors can clarify this, I will change my scores.

**Main Review:**

**Post Rebuttal**

I have seen the author's response. Overall, I still think the author's response does not fully address my first question (why not use QNN if the neuromorphic HW can support up to 8-bit spikes?). Their responses referring that 1) SNN uses binary activation, 2) multiplication-free computation and 3) higher adversarial robustness all are the property of conventional SNN rather than the multi-bit SNN. However, I also acknowledge that the authors made adequate efforts for rebuttal, which I indeed appreciate. I will increase my score to 6 and I hope authors can rigorously discuss the point of this multi-bit spike in the next version.

PS: According to your baseline paper, the tdBN-Spiking-ResNet-50 has 65% accuracy while your result is 58%?

-----


Overall, the authors propose a reasonable architecture to address the residual mapping issue in the spiking networks. However, the overall algorithm seems to be a minor solution, as the only difference is the swap of two operations. This work main experiment with  ImageNet experiments, which I appreciate.


**Pros**:

+ The topic of this paper is interesting, the residual path is an important technique in ANN, and I am happy to see a paper study this architecture in SNN.

+ The experiments in ImageNet show improvements.

**Cons**:

- My biggest concern of this work is the non-binary spikes in the network. Although the authors propose two variant element-wise ops: namely AND and IAND, they finally adopt the ADD operation, which means the spikes after the operation is not binarized. And the bit-width that is used to represent this spike will continue to increase. This seems to be an undesired characteristic in SNN since the neuromorphic hardware operates with binary activation. Thus the energy consumption may be significantly higher than the spiking resnet.

- In my view, the spiking resnet is more biology-plausible. Is there any biological evidence that two spikes can be fused together?

**Time Spent Reviewing:**

2

---

> ### Author Response · Authors · 2021-08-09
> **Response to Reviewer 2Q6H**
>
> Thank you for your very detailed comments and suggestions for improvement. We would like to address your concerns and answer your questions in the following.
>
> > My biggest concern of this work is the non-binary spikes in the network.
>
> Thanks for your suggestion. We agree that using ADD operation will cause non-binary outputs. However, we think it is not a problem for neuromorphic implementation because most neuromorphic chips handle a few bits for spike gradation. For example, the Tianjic chip [Pei et al., 2019] supports 8-bits spike gradation, and the Loihi [Davies et al., 2018; Davies et al., 2021] supports no more than 32 bits graded-values, which depends on the size of the addresses and the number of neurons.
>
> [Pei et al., 2019] Jing Pei, et al. Towards artificial general intelligence with hybrid Tianjic chip architecture. Nature. 2019, 572(7767):106-11.
>
> [Davies et al., 2018] Davies, Mike, et al. Loihi: A neuromorphic manycore processor with on-chip learning. IEEE Micro. 2018, 38(1):82-99.
>
> [Davies et al., 2021] Davies, Mike, et al. Advancing neuromorphic computing with Loihi: A survey of results and outlook. Proceedings of the IEEE. 2021, 109(5):911-934.
>
> > And the bit-width that is used to represent this spike will continue to increase.
>
> We agree the spikes accumulation between two downsample blocks will grow with the depth, but it grows slowly. As the maximum sequential blocks number for ResNet 18/34/50/101/152 structure is 2/6/6/23/36 (the *conv4\_x* module in Ref.[12]), the maximum possible element in $O^{l}$ is 3/7/7/24/37. In the experiment, we also check the distribution of $O^{l}_{j} [t]$ in SEW ResNet-152. As shown in Tab.R4, the maximum spike accumulation in SEW ResNet-152 is 18, which can be represented with 5 bits. Considering that the ratio of large spike accumulation ($O^{l}\_{j}[t]>16$) is less than $10^{-11}$, we believe that ignoring them and using fewer bits will not cause too much performance drop, which can be done by adding an output saturation to the ADD operation.
>
> | SEW ResNet/$O^{l}\_{j}[t]$ | 0        | 1        | 2        | 3        | 4        | 5        | 6        | 7        | 8        | 9        | 10       | 11       | 12       | 13       | 14       | 15       | 16       | 17       | 18       |
> | -------------------------- | -------- | -------- | -------- | -------- | -------- | -------- | -------- | -------- | -------- | -------- | -------- | -------- | -------- | -------- | -------- | -------- | -------- | -------- | -------- |
> | 18                         | 7.80E-01 | 1.99E-01 | 2.00E-02 | 5.52E-04 |          |          |          |          |          |          |          |          |          |          |          |          |          |          |          |
> | 34                         | 6.21E-01 | 3.13E-01 | 6.13E-02 | 4.38E-03 | 2.10E-04 | 1.15E-05 | 1.66E-06 | 1.62E-09 |          |          |          |          |          |          |          |          |          |          |          |
> | 50                         | 5.49E-01 | 3.95E-01 | 5.24E-02 | 3.61E-03 | 1.81E-04 | 7.20E-06 | 5.14E-08 | 2.21E-10 |          |          |          |          |          |          |          |          |          |          |          |
> | 101                        | 3.78E-01 | 2.86E-01 | 1.24E-01 | 9.11E-02 | 6.98E-02 | 3.60E-02 | 1.21E-02 | 2.98E-03 | 4.67E-04 | 4.39E-05 | 4.18E-06 | 1.58E-07 | 3.07E-10 | 6.72E-12 |          |          |          |          |          |
> | 152                        | 3.03E-01 | 2.58E-01 | 1.09E-01 | 7.93E-02 | 7.76E-02 | 6.70E-02 | 5.13E-02 | 3.29E-02 | 1.56E-02 | 5.57E-03 | 1.47E-03 | 2.90E-04 | 4.65E-05 | 5.04E-06 | 3.90E-07 | 1.44E-08 | 2.45E-10 | 3.04E-12 | 3.04E-12 |
>
> **Table R4: The distribution of $O^{l}\_{j}[t]$.**
>
> For those neuromorphic chips that only support 1-bit spikes, changing computing order can be a solution for ADD operation. We can find that all outputs of the ADD operation are sent to convolutional or fully connected layers, which are linear, indicating that add before linear operation can be changed to linear operation before add. More specifically, we can change *ADD* to *Concatenate On Batch Dimension*, send concatenated spikes to the convolutional or fully connected layer, sum the outputs, and we will get the same results. Note that if the linear layer is biased, the bias should be scaled according to the input's number.
>
> We have added the analysis of energy consumption according to the method Reviewer 96mQ suggested.  The proposed SEW ADD ResNet can reach 137-210 times energy efficiency than ANN. Please refers to Tab.R5.
>
> > In my view, the spiking resnet is more biology-plausible. Is there any biological evidence that two spikes can be fused together?
>
> Our model is bio-inspired, but not biologically plausible. The possible biological implementations are as follows. When using ADD in the SEW ResNet, the situation is similar to the last question. The addition of two spikes can be implemented by two input spikes delivered to different dendrites of a neuron.
> When using AND and IAND, $g$ plays a role similar to gating, which can be implemented by one interneuron that transiently inhibits a second interneuron, allowing increased activity in the excitatory targets of the latter (see Fig. 1 of [Oliver and Beck, 2018]).
>
> [Oliver and Beck, 2018] Braganza Oliver, Heinz Beck. The circuit motif as a conceptual tool for multilevel neuroscience. Trends in Neurosciences. 2018, 41(3): 128-136.

---

> > ### Comment · Reviewer_2Q6H · 2021-08-12
> > **Thanks for the Rebuttal, Some Further Questions Remain**
> >
> > I would appreciate the authors' effort for the rebuttal. I would like to ask more questions. If the neuromorphic implementation can handle 8-bit integer values, then what is the effect of SNN? Why not using quantization neural networks for efficient inference since an 8-bit ResNet-18 has full precision accuracy (70%), which is even higher than SEW-ResNet-152.
> >
> > Besides, I appreciate the complexity tabular visualization. Please explain how to compute the energy of multi-bit spikes. How to do convolution between multi-bit spikes and 32-bit weights?

---

> > > ### Author Response · Authors · 2021-08-20
> > > **[2] Response to Reviewer To 2Q6H**
> > >
> > > > Please explain how to compute the energy of multi-bit spikes. How to do convolution between multi-bit spikes and 32-bit weights?
> > >
> > > We use the same calculation method from Ref.[15] according to Reviewer 96mQ's suggestion. Thanks for your question! We find that there exists a problem with this method for multi-bit spikes.  We utilize 77fJ/SOP for SNN as the power consumption baseline, which is reported from the ROLLS neuromorphic processor [Qiao et al., 2015]. We find that this chip does not support multi-bit spikes. The Tianjic chip can support multi-bit spikes. However, they have not collected the power consumption data for multi-bit spikes (we have asked the authors of Tianjic chip for data).
> > >
> > > To calculate the power consumption correctly, we trained the SEW IAND ResNet on ImageNet. The test accuracy is shown in Tab.R6, and the power consumption is shown in Tab.R7.
> > >
> > > | Network   | SEW ResNet(ADD) |          | SEW ResNet(IAND) |          | Spiking ResNet |          |
> > > | --------- | --------------- | -------- | ---------------- | -------- | -------------- | -------- |
> > > |           | Acc@1(%)        | Acc@5(%) | Acc@1(%)         | Acc@5(%) | Acc@1(%)       | Acc@5(%) |
> > > | ResNet-18 | 63.18           | 84.53    | 61.71            | 83.48    | 62.32          | 84.05    |
> > > | ResNet-34 | 67.04           | 87.25    | 64.76            | 85.95    | 61.86          | 83.69    |
> > > | ResNet-50 | 67.78           | 87.52    | 66.20            | 86.64    | 57.66          | 80.43    |
> > >
> > > **Table R6: Test accuracy of SEW ADD/IAND ResNet and Spiking ResNet on ImageNet.**
> > >
> > > | ResNet Structure | 18     | 34     | 50     |
> > > | ---------------- | ------ | ------ | ------ |
> > > | ANN OP (GFLOP)   | 1.82   | 3.68   | 4.14   |
> > > | SNN OP (GSOP)    | 1.61   | 3.15   | 3.24   |
> > > | ANN Power (mJ)   | 22.75  | 46     | 51.75  |
> > > | SNN Power (mJ)   | 0.12   | 0.24   | 0.25   |
> > > | A/S Power Ratio  | 183.51 | 189.65 | 207.43 |
> > >
> > > **Table R7: Comparison of power consumption of SEW IAND ResNet and ResNet.**
> > >
> > > [Qiao et al., 2015] Ning, Qiao, et al. A reconfigurable on-line learning spiking neuromorphic processor comprising 256 neurons and 128K synapses. Frontiers in neuroscience. 2015, 9: 141.
> > >
> > > > Why not using quantization neural networks for efficient inference since an 8-bit ResNet-18 has full precision accuracy (70%), which is even higher than SEW-ResNet-152.
> > >
> > > We agree that the classification performance of SNNs is slightly worse than quantization neural networks. However, SNN has its own distinctive properties, which have increasingly aroused researchers’ great interest in recent years.We think SNNs have three main advantages:
> > >
> > > 1) SNNs use binary activations, which are more efficient than 8-bit ones, because we do not need to do multiplications. Instead, we just add the synaptic weight to the potential when there is an input spike (Accumulate operation vs. Multiply-Accumulate operation).
> > >
> > > 2) Due to the event-driven calculation, sparse activation, and multiplication-free characteristics, the existing neuromorphic chips have shown that SNNs have greater energy efficiency than ANN.
> > >
> > > 3) SNNs have inherent adversarial robustness. The adversarial accuracy of SNNs under gradient-based attacks is higher than ANNs with the same structure [Sharmin et al., 2020].
> > >
> > > [Sharmin et al., 2020] Saima Sharmin, et al. Inherent adversarial robustness of deep spiking neural networks: Effects of discrete input encoding and non-linear activations. European Conference on Computer Vision (ECCV), 2020.

---

### Author Response · Authors · 2021-08-09
**To All Reviewers**

We thank all reviewers for their time and insightful feedback. In this general response, we would like to address the concern about performance on CIFAR10-DVS and the comparison of SEW ResNet-152 and Spiking ResNet-152.

**Performance on CIFAR10-DVS**

*2021.9.7: We find a bug in our codes of training on CIFAR10-DVS and the previous accuracy of $T=4$ is overestimated. Now we correct the experiments results. The conclusion is the same as we still get higher accuracy with smaller time-steps than Spiking ResNet.*

We have evaluated the performance of SEW ResNet on the CIFAR10-DVS dataset, which is obtained by recording the moving images of the CIFAR-10 dataset on an LCD monitor with a DVS camera. As the CIFAR10-DVS dataset is more complicated than the DVS Gesture dataset, we use the network structure named Wide-7B-Net, which is similar to 7B-Net but with more channels. The structure of Wide-7B-Net is {c64k3s1-BN-PLIF-{SEW Block(c64)-MPk2s2}\*4-c128k3s1-BN-PLIF-{SEW Block(c128)-MPk2s2}\*3-FC10}. Here c64k3s1 means the convolutional layer with channels 64, kernel size 3, stride 1. MPk2s2 is the max pooling with kernel size 2, stride 2. The symbol {}\*4 denotes four repeated structures.

| Network                                                      | Accuracy(%)      | Parameters | $T$      |
| ------------------------------------------------------------ | ---------------- | ---------- | -------- |
| c64k3s1-BN-PLIF-{SEW Block(c64)-MPk2s2}\*4-c128k3s1-BN-PLIF-{SEW Block(c128)-MPk2s2}\*3-FC10 (Wide-7B-Net) | 64.8, 70.2, 74.4 | 1.19M      | 4, 8, 16 |
| {c128k3s1-BN-PLIF-MPk2s2}*4-DP-FC512-PLIF-DP-FC100-PLIF-APk10s10 [6] | 74.8             | 17.4M      | 20       |
| Spiking ResNet-19 with td-BN [56]                            | 67.8             | 11.18M     | 10       |

**Table R1: Comparison with the state-of-the-art (SOTA) methods on the CIFAR10-DVS dataset.**

In Tab.R1, we compare SEW ResNet with the previous Spiking ResNet. One can find that our method achieves better performance (70.2% v.s. 67.8%) and fewer time-steps  (8 v.s. 10) than the Spiking ResNet [56]. We also compare our method with the state-of-the-art (SOTA) supervised learning methods on CIFAR10-DVS. The accuracy of our Wide-7B-Net is slightly lower than the current SOTA method [6] (74.4% v.s. 74.8%), which uses 1.25 times as many simulation time-steps $T$ (16 v.s. 20) and 14.6 times as many the number of parameters  (1.19M  v.s. 17.4M). Moreover, when reducing $T$ shapely to $T=4$, our Wide-7B-Net can still get the accuracy of 64.8%.

| Dataset     | Learning Rate Scheduler        | Epoch | $lr$ | Batch Size | $T$  | $n_{gpu}$ |
| ----------- | ------------------------------ | ----- | ---- | ---------- | ---- | --------- |
| CIFAR10-DVS | Cosine Annealing, $T_{max}=64$ | 64    | 0.01 | 16         | 4,8,16 | 1         |

**Table R2: Hyper-parameters of the Wide-7B-Net for CIFAR10-DVS datasets.**

Tab.R2 shows the hyper-parameters of the Wide-7B-Net for the CIFAR10-DVS dataset. We use the same AER data pre-processing method as the DVS128 Gesture dataset. We do not use *random temporal delete* because the CIFAR10-DVS dataset is obtained with static images.

**Comparison of SEW ResNet-152 and Spiking ResNet-152**

We have added the results of ResNet-101/152 structure in Tab.R3. One can find that the Spiking ResNet suffers the degradation problem. As the layer increases, the test accuracy decreases. For example, the test accuracy of Spiking ResNet-101 and Spiking ResNet-152 on Imagenet are only 31.79% and 10.03%, respectively. In contrast, the deeper SEW ResNet has higher test accuracy than the shallower SEW ResNet.

| Network    | SEW ResNet |          | Spiking ResNet |          |
| ---------- | ---------- | -------- | -------------- | -------- |
|            | Acc@1(%)   | Acc@5(%) | Acc@1(%)       | Acc@5(%) |
| ResNet-18  | 63.18      | 84.53    | 62.32          | 84.05    |
| ResNet-34  | 67.04      | 87.25    | 61.86          | 83.69    |
| ResNet-50  | 67.78      | 87.52    | 57.66          | 80.43    |
| ResNet-101 | 68.76      | 88.25    | 31.79          | 54.91    |
| ResNet-152 | 69.26      | 88.57    | 10.03          | 23.57    |

**Table R3: Test accuracy on ImageNet.**

---

### Decision · Program_Chairs · 2021-09-27

**Decision:**

Accept (Poster)

**Comment:**

By allowing multi-bit spiking, a novel residual block structure is proposed for neuromorphic deep learning. Thorough experiments were run to compare performance and power consumption. Reviewers generally agreed that this contribution is interesting for the spiking neural network community. They also agreed that the writing can be improved. Please make sure to include the results included in the rebuttal in the main paper as well.